# Artificial Intelligence in Orthopedic Radiography Analysis: A Narrative Review

**DOI:** 10.3390/diagnostics12092235

**Published:** 2022-09-16

**Authors:** Kenneth Chen, Christoph Stotter, Thomas Klestil, Stefan Nehrer

**Affiliations:** 1Department for Orthopedics and Traumatology, Landesklinikum Baden-Mödling, 2340 Mödling, Austria; 2Department for Health Sciences, Medicine and Research, Danube University Krems, 3500 Krems, Austria

**Keywords:** musculoskeletal imaging, artificial intelligence, machine learning, deep learning, radiograph, X-ray

## Abstract

Artificial intelligence (AI) in medicine is a rapidly growing field. In orthopedics, the clinical implementations of AI have not yet reached their full potential. Deep learning algorithms have shown promising results in computed radiographs for fracture detection, classification of OA, bone age, as well as automated measurements of the lower extremities. Studies investigating the performance of AI compared to trained human readers often show equal or better results, although human validation is indispensable at the current standards. The objective of this narrative review is to give an overview of AI in medicine and summarize the current applications of AI in orthopedic radiography imaging. Due to the different AI software and study design, it is difficult to find a clear structure in this field. To produce more homogeneous studies, open-source access to AI software codes and a consensus on study design should be aimed for.

## 1. Introduction

Artificial intelligence (AI) is a growing field that has experienced exponential growth, especially in the past five years (Figure 1). Reasons for that include the high-performance central- and graphics-processing units (CPU and GPU), the availability of huge amounts of data, and the development of learning algorithms [1].

As early as 1955, John McCarthy et al. proposed a research project on AI to investigate their conjecture that every aspect of intelligence could be reproduced by a machine [2]; thus, the field of AI was founded, which became the foundation of subsequent computer research and development [3]. The early success of AI was due to the fact that tasks that were easy to formally program but difficult for humans to perform could be easily solved by AI. Ironically, tasks that seem easy for humans, like the recognition of objects or speech, are much harder to solve because they are done intuitively and are therefore difficult to put in a formal way such as code [4].

In the age of digitalization, where AI is applied in many fields of daily life, medicine is a field that has yet to make up the leeway. Although AI applications in medicine are increasing, most X-ray images are still assessed manually with narrative image descriptions [4,5,6]. AI in healthcare has the potential to create new methods to reduce the steadily increasing workload of clinicians, while also improving research and development, and recognizing underlying patterns that otherwise could not be recognized [7]. As there is always a risk of error when diagnosing radiographs, manually assessing them under severe and urgent conditions can elevate that risk to 40% or more [8]. As computers are not affected by their surroundings or stress, the performance of AI stays more consistent.

This narrative review gives an overview of AI in medicine, discusses the challenges, and summarizes the current applications of AI in orthopedic radiography analysis (Figure 2). 

### 1.1. Inter- and Intra-Observer Variability

When manually measuring or classifying radiographs, aside from being prone to error under certain conditions, there is also the aspect of inter- and intraobserver variability. Inter- and intraobserver variability in orthopedic radiology can occur everywhere, for instance when measuring the acetabular component after hip arthroplasty [5] or assessing the sacroiliac joint for sacroiliitis [6]. While there have been studies comparing inter- and intraobserver variability, Rutgers and colleagues stated that although the quality of diagnosis by inexperienced clinicians can be improved through training, supervision by experienced observers is necessary [8]. It has been shown that it is possible to improve the variability and difference in quality through a computer-aided diagnosis, which is not exclusively found in orthopedic radiology, but also other fields of medicine [9].

While specialties such as oncology, cardiology, and neurosurgery have implemented AI in their daily practice, orthopedics and orthopedic surgery are only slowly embracing it, although its application is growing [10,11,12]. In 2017, Cabitza et al. could already see increasing demand for systematic reviews in the field of AI in orthopedics. Figure 1 demonstrates that the number of papers published has drastically increased over the last few years [13].

### 1.2. Artificial Intelligence and Machine Learning

Machine learning (ML) is part of the field of AI (Figure 3), which is based on algorithms that describe the relationships between variables. In conventional statistical analyses, the goal is to find inferences by understanding the relationships between variables to reach conclusions or make predictions, based on the collected data. Data that is collected through the conversion of medical images, a process called radiomics, contains information on underlying pathophysiologies that can only be revealed by quantitative image analysis [14]. One of the advantages of ML is that it can formulate hypotheses based on the collected data, with only occasional input from humans, making it flexible. That flexibility, however, can also be a weakness [15]. While the primary goal of ML is to make predictions, in contrast to linear or logistic regression, the accuracy of the prediction or the “what” is more important than understanding the way or the “how” of reaching it, e.g., understanding the relationships of pixels of an X-ray is not important as long as the diagnosis of a fracture is correct [16]. Although this “black box” nature of ML models potentially offers more accurate performance in prediction, it comes with the cost of less understanding of the impact that variables might have on the outcome, making it more difficult to pinpoint the exact problems needed to treat [17].

The reason the above-mentioned flexibility of an ML model can be a weakness is overfitting. Overfitting is one major problem when it comes to ML. The term overfitting describes an error made by an ML model when it mistakenly extracts parameters out of a training dataset and assigns more importance to that parameter than it has in reality.

Typically, an ML algorithm runs the input data through its layers to get an output. The output is then compared with a “ground truth” (e.g., results of human readers as the ground truth for ML radiography analysis) and the parameters and their importance are adjusted repeatedly to reach an output with a minimal difference from the ground truth. This favours the parameters that have more “weight” in the decision-making for the output. If the ML model runs through the same dataset too many times, overfitting happens, and residual variation or “noise” are used for the predictions. One solution to overcome the overfitting problem is to collect bigger datasets for training. Although different amounts of data are required, depending on the study, a basic starting point of 1000 images per class will have a good chance for high accuracy. For medical data, it is often difficult to collect such large numbers due to confidentiality issues and high costs in acquiring the ground truth. To achieve better generalizability and reduce the risk of overfitting, external validation is necessary [18].

ML is broadly divided into three categories: supervised learning, unsupervised learning, and reinforcement learning [16,19,20].

Supervised learning is the most common form of learning used in medical practice. Typically, a supervised ML model is “trained” and validated on a dataset with given inputs and outcomes. After successful training and validation, the model should be able to make predictions on new data. Those predictions can be either binary or continuous, which is then referred to as a classification algorithm or segmentation algorithm, respectively [16,20]. An example of a classification algorithm would be fracture detection, where there either is (true) or is no (false) fracture. A segmentation algorithm might be used to predict life expectancy for non-treated femur fractures.

In contrast to supervised learning, in unsupervised learning the outcome is not put in manually. The goal here is to recognize patterns in a dataset, such as latent subspaces, by comparing the values of variables. If predefined variables show similar values, they are grouped together building a so-called cluster. By performing clusterization (Figure 4), patterns that would otherwise be difficult to recognize can be discovered [13,16,20,21]. 

Reinforcement learning is the third category of ML, with the goal of optimizing the process for a certain reward. For example, in a game of chess or poker [22], the individual steps and sequences would be difficult to compute but the outcome of winning or losing can be predefined as a reward, and the algorithm learns the most optimal way of reaching it [20].

The most successful learning models recently developed are artificial neural networks (ANN) and deep learning (DL) models. Neural networks are learning models with an architecture similar to the mammalian cortex, which consists of two visible and a variable number of hidden layers. ANNs, with a high number of hidden layers, are considered DL models [16,20]. There is no clear cut-off of number of layers needed for a neural network to be defined as a DL model. Particularly suited for perceptual tasks, such as object recognition, are DL models that resemble the human visual cortex called convolutional neural networks (CNNs). The CNNs contain special types of layers called convolutional layers that apply filters on the input and pooling layers, which compute and filter the preceding layer, resulting in less computational load and a lower risk of overfitting (Figure 5). Today, CNNs are the majority of neural network architectures used in computer vision [20].

### 1.3. Applications of AI in Musculoskeletal Radiology

Musculoskeletal (MSK) diseases are the most common work-related health issues in the European Union (EU). According to the European Agency for Safety and Health at work, three out of five people have backache or pain in the neck or upper limbs [23]. In the “Global estimates for the need of rehabilitation…” study, MSK disorders contributed most to the need for rehabilitation, with lower back pain being the most prevalent condition in more than half of the countries analyzed [24]. In addition, MSK trauma is among the most common reasons for unscheduled visits to the emergency department [25,26]. Missed or delayed diagnosis can lead to an increase in morbidity and mortality. With increasing numbers of images, physicians are pressured to work faster whilst maintaining quality standards. It has been estimated that the rate of radiologic error ranges from 4% to around 30% depending on the study sample with the two most common errors of underreading (a finding is present on the image but is missed) and satisfaction of search (findings missed because the search was not continued after one abnormality was found) [27]. When the 4% is extrapolated on one billion radiologic images a year, an estimated forty million radiologic errors are made [25]. Those errors might result in patients receiving inadequate treatment and the number of errors could be reduced if AI would be applied as a tool for clinicians to reduce workload [19].

## 2. Material and Methods

A PubMed search was performed on 5 May 2022 containing the following search string: (“Artificial Intelligence”[Title/Abstract] OR “deep learning”[Title/Abstract] OR “neural network”[Title/Abstract] OR “machine learning”[Title/Abstract]) AND (“musculoskeletal”[Title/Abstract] OR orthopedic*[Title/Abstract] OR orthopaedic*[Title/Abstract] OR muscle*[Title/Abstract] OR skeletal*[Title/Abstract] OR sceletal*) AND (radiograph*[Title/Abstract] OR “X-Ray”[Title/Abstract] OR “X-Rays”[Title/Abstract]). Additionally, reference lists of framework papers were searched.

### 2.1. AI in X-ray Imaging

#### 2.1.1. Fracture Detection

There have been a number of studies with different products on fracture detection using AI on plain radiographs [28,29,30,31,32,33,34]. Fractures are the leading type of missed diagnosis [35]. DL models can classify images to detect whether there is a fracture or not and then use segmentation to assess localization and fracture type [36]. Yu and colleagues developed and tested a DL algorithm and evaluated its accuracy in the detection and localization of hip fractures. They found a sensitivity and specificity of 97.1% and 96.7%, respectively, for binary classification. In fracture localization, however, performance was lower with the sensitivity ranging from 95.8% in normal findings, 84.1% in subcapital/transcervical fractures, 76% in basicervical/intertrochanteric fractures, to 20% in subtrochanteric fractures. Although, when compared to subspecialty musculoskeletal radiologists the performance was lower, the DL algorithm showed high accuracy in fracture detection [33].

Liu and colleagues trained a DL algorithm for tibial plateau fractures (TPF) in radiographs and compared its performance with orthopedic physicians. The algorithm automatically labelled the suspicious TPF region with a rectangle. The accuracy of the algorithm was comparable to human performance with 0.91 and 0.92 ± 0.03, respectively. There was, however, a significant difference in the time spent assessing the X-rays, with the AI being 16-times faster than orthopedic physicians. The authors state that the working conditions for orthopedic physicians were not ideal to simulate real working conditions, since the study was conducted in nonemergency and under time-free constraints on the diagnosis process [29].

Missed diagnoses of vertebral fractures in plain radiographs can be as much as 30% [37]. Murata and colleagues developed a CNN to detect vertebral fractures and compared it to orthopedic residents, orthopedic surgeons, and spine surgeons. They found that there was no significant difference between the CNN, the orthopedic residents, and surgeons but spine surgeons had significantly higher accuracy, sensitivity, and specificity [30].

AI has been shown to be on par with or even outperform expert radiologists in fracture detection on radiographs and is a potential tool to aid clinicians to minimize errors and reduce burnout [25].

#### 2.1.2. Classification

DL models can not only be trained to detect fractures but also for segmentation and classification. In knee osteoarthritis (OA), the measurement of joint space width (JSW)–which is the gap between the femur and tibia–and an indirect measure of cartilage width is the only recommended imaging biomarker as a structural endpoint in clinical trials, as recommended by the United States Food and Drug Administration (FDA) [38]. Automated knee segmentation has been investigated for accurate identification of the femur and tibia by several studies [39], and has been shown to deliver a more standardized and objective classification of joint-space narrowing [38]. The most widely used grading system for OA, however, is the Kellgren-Lawrence (KL) scale. The KL scale is recognized by the World Health Organization as the standard for clinical studies and includes radiological features: osteophytes, subchondral sclerosis, pseudocystic areas, and narrowing of the joint space [40]. Since several studies reported that the KL grading system suffers from high subjectivity and interobserver variability, studies investigating the benefit of AI-aided classification found that the consistency between physicians increased when automated knee OA software was used [41].

Since diagnosis and treatment in pediatric orthopedic disorders can depend on bone age, although a time-consuming activity, an accurate assessment is necessary. Bone age is closely related to skeletal maturity, whereas chronological age is based on an individual’s date of birth. The two most common methods to assess bone age are the Greulich-Pyle and Tanner-Whitehouse methods. Both methods analyze the epi- and diaphyses of hand radiographs and compare them to reference images, leaving both methods prone to inter-and intraobserver variability [42]. Larson and colleagues tested a DL bone age assessment model on 14,036 hand radiographs and found a comparable performance of the DL model to trained human reviewers, with a difference between bone age estimates of 0 years, root mean square of 0.63 years, and a mean absolute difference of 0.50 years [43]. It is important, however, that caution is taken when assessing bone age in patients with skeletal disorders like Hutchinson-Gilford progeria syndrome, where patients show a more advanced rate of skeletal maturation [44]. Large datasets that are needed to train ML models are much more difficult to acquire for rare diseases, which makes experience and supervision by humans a necessity.

The standard tool used for the diagnosis of osteoporosis is the assessment of bone mineral density (BMD). Dual-energy X-ray Absorptiometry (DXA) is the gold standard of methods used for the assessment of BMD and is the basis for the WHO guidelines for the diagnosis of osteoporosis [45]. Yamamoto and colleagues investigated a DL model to diagnose osteoporosis in hip radiographs. They tested five different CNNs and found high accuracy. When adding routinely available patient variables such as age, gender, and BMI, the diagnostic accuracy improved to an AUC of 0.906 with a ResNet50 CNN architecture [46,47].

Total joint replacement (TJR) is the most common elective surgical procedure for end-stage degenerative joint disease. The primary diagnosis leading to TJR is OA accounting for up to 81% of hip and 94% of knee replacements [48]. Implants fail within 15–20 years in around 10% of all TJRs and revision surgery is needed. The planning for revision surgery requires identification of the implant’s model, which can be difficult when the primary surgery was performed more than 10 years ago and/or in a different country. Studies have shown that DL models can automatically identify implant models in the hip as well as shoulders on plain radiographs [49,50]. Borjali and colleagues compared the performance of a DL model to that of orthopedic surgeons and found similar or higher performance by the DL model. They also found that the DL model was significantly faster than surgeons, taking about 0.06 s and 8.4 min, respectively, given that the surgeon did not know the implant model by experience [49].

### 2.2. Measurements

Another application for AI in radiographs is in automated measurements, for example, leg alignment, joint orientation, leg length, or implant orientation [51,52,53,54]. Long-leg radiographs (LLR) are routinely acquired and manually measured to assess malalignment, which is considered a major contributing factor in OA [53]. The measurement of leg-length discrepancies (LLD), hip-knee-ankle angle (HKAA), or femoral anatomic-mechanical angle (AMA) is relatively simple but is a time-consuming task [55]. Several studies have tested DL models on automatic measurements of the lower limb measuring HKA, AMA, and LLD. The studies found that the AI was non-inferior when compared to human readers (accuracy ranging from 89.2 to 93%) but was significantly faster [52,53,54]. However, severe deformities of the lower limb and bad image quality are common exclusion criteria for the studies mentioned above, emphasizing again that experienced clinicians are indispensable as supervisors. This becomes even more important in undiagnosed bony abnormalities.

### 2.3. Computed Tomography and Magnetic Resonance Imaging

Apart from radiographs, studies on AI in MSK computed tomography (CT) have also been conducted. For example, for the detection and classification of fractures of the calcaneus, femur, and mandibula AI could be a useful tool for radiologists and orthopedic surgeons to reduce workload and improve accuracy [56,57,58,59]. When assessing rib fractures in CTs, radiologists achieved higher accuracy while reducing their reading time with the assistance of a DL model compared to without [60]. Aside from the analysis of bones, AI can also be used in the analysis of skeletal muscle and confirmation of atrophy in CT scans [61].

Magnetic resonance imaging (MRI) is unparalleled in detecting musculoskeletal diseases and is the preferred method for diagnosing knee injuries [62,63]. Several studies have applied AI in the assessment of MRI, for example, in the detection of injuries of the anterior cruciate ligament (ACL) [57], meniscus tears [64], or cartilage lesions [65]. Machine learning in MRI can also be used for the assessment of vertebrae, discs, and muscles in patients with lower back pain [66]. In a systematic review, Siouras and colleagues stated that AI in MRI has the potential to be on par with human-level performance and has shown a prediction accuracy ranging from 72.5–100% [67].

Research has shown that orthopedic problems can also be related to orthodontics. Neural networks have produced a good performance in treatment planning and can offer guidance for less experienced orthodontists [68]. For example, malocclusion has been shown to be related to scoliosis and weak body posture [69]. The implementation of MRI in orthodontics has shown great potential for radiation-free imaging. Compared to the currently used cone-beam CT, MRI has shown better results in the detection of periodontal structures, soft tissue debris, and unfilled spaces in radicular canals, which is necessary information for endodontic treatment [70,71,72]. Additionally, orthodontic treatment is mostly received during childhood, which makes MRI even more desirable [73]. As mentioned before, overfitting is a challenge for ML algorithms. The size and quality of data, especially with MRI images where the signal-noise ratio can be worse, determine and influence the performance quality. Although MRI images show a lot of information, they also have more noise [74].

### 2.4. Internal and External Validation

The development of ML models requires training and validation, where validation is the process of evaluating the accuracy of the model. Generally, the validation processes can be divided into internal and external validation. Internal validation splits the same dataset to train and validate the ML model. The process where one part of the data is used for training and the rest is used for validation repeatedly, until all parts of the dataset have been used for training, is referred to as cross-validation. An advantage of using internal validation is that it is easier and more economical to perform than external validation since only one dataset is needed. The disadvantage, however, is that even if cross-validation shows high accuracy, it does not give information about the reproducibility or meaningfulness of findings. Ho and colleagues made a comparison between the learning of ML models and humans: When two humans study for an exam and one person critically evaluates and learns to understand the subject and the other person simply memorizes, both can reach a result of 99%. This result, however, does not give us information about the performance of those two humans on another exam of the same subject with different questions. The same principle can be applied to ML models: two models that are trained on the same dataset can reach the same result of 99% accuracy but the way the ML model learned to reach that accuracy may differ, which can lead to different results in performance on other datasets [75]. Additionally, the risk of bias in internally validated models is increased, because the training dataset may not represent the target population properly. Zech and colleagues noticed that the word “portable”, which was a label indicating that X-rays were taken with a portable scanner, was detected by the ML model and used as an indicator for a higher possibility of disease [76]. While that may have been the case in that specific hospital setting, the ML model suffered from overfitting because the reproducibility for different hospitals with different imaging procedures, that might not have “portable” written on the radiographs, can differ in performance.

External validation, on the other hand, uses one dataset for training and a different (external) dataset for validation. The validation dataset is supposed to be profoundly different from the training dataset, e.g., in a different geographical location. Although collecting a different dataset requires more effort, external validation can be used to counteract the above-mentioned disadvantages of internal validation and is considered important evidence for generalizability (Figure 6) [77]. 

Since internally validated ML models tend to perform better on the data they were trained on [78,79], Oliveira and colleagues investigated ML models for fracture detection that were geographically (datasets from different locations) or temporally (datasets from different time points) externally validated. Out of 36 CNNs that were investigated, only four (11%) used a form of external validation, which limits the potential to use those CNNs in clinical practice [80].

## 3. Discussion

This narrative review gives an introduction to AI and summarizes the current applications of AI in orthopedic radiography imaging. The literature on AI in orthopedics is rapidly growing with different types of studies being conducted. Due to that heterogeneity, it is difficult to structure and compare data properly. AI in orthopedic radiographs is applied in the fields such as fracture detection, identification and classification of bone age, osteoporosis or implants, and measurement of alignment or implant position. Studies mostly show non-inferior performance when compared to physicians, orthopedic surgeons, and radiologists, whilst being faster at performing the tasks. For some, this now might raise the question of whether AI will replace clinicians, especially radiologists. When ML first had its hype in 2016, some authors hypothesized that the tasks of radiologists could be entirely replaced by AI, and some even stated that the training of radiologists should be stopped. This narrative, however, has been revised by many experts in the field that are working together with radiologists. Currently, AI is often trained on a single disease or a small set of diseases, which is far under the scope of what radiologists do. The performance of AI can reach that of a subspecialty thoracic radiologist when it comes to the diagnosis of common chest conditions; however, in contrast to the human radiologist, AI falls behind in detecting uncommon diseases. Hence, the very fitting comparison by Langlotz: “AI is impressive in identifying horses but is a long way from recognizing zebras” [81].

The performance of clinicians when aided by AI software has been shown to improve in comparison to AI or human performance alone. This could not only be applied in clinical practice to improve diagnosis quality but also as a training tool, especially for junior doctors with less experience [41].

Since medicine is in the process of digitalization, it is also important to familiarize junior doctors with the topic of AI for clinical practice. As Çalışkan and colleagues state, medical students need to understand why AI can support doctors in clinical practice [82]. On the other hand, we cannot rule out the possibility that medical students and junior doctors could rely too much on AI, which can become a hindrance in developing analytical and observational skills as well as in learning the process of decision-making.

The importance of external validation has been explained above. As Oliveira and colleagues point out, external validation, which means validating a DL model on a different dataset than it was trained on, is hardly carried out and was only reported in 11% of studies investigating CNNs for fracture detection. To achieve a higher quality standard and reduce the risk of bias, geographically external validation should always be performed to assess performance and generalizability [80].

Another factor that is rarely investigated is ethnicity. In a systematic review, Dallora and colleagues stated that only two out of twenty-six studies considered ethnicity as a factor, and found that studies are predominantly conducted in the United States and Western Europe whilst it is known that, e.g., skeletal maturation, occurs earlier in Indonesians and later in African Americans when compared to Caucasians [42].

The application of AI models in orthopedics focused on predicting outcomes is still in its infancy. The studies investigating prediction models are mostly focused on medical management (predictions of delayed discharge or readmission) and the subspecialty of spine surgery [83]. We believe that there should be more attention given to AI prediction models in the future since providing predictions is a task that cannot be performed by humans. Currently, the predictions made by clinicians can only be made based on personal experience and statistical incidences; however, if the prediction of diseases could be made accurately, it could affect the following treatment.

The term big data was invented by computer scientists to describe the evolving technology based on digitally collected and rapidly expanding data. The idea of big data in healthcare is to share data between practitioners and patients to improve healthcare delivery [84]. Currently, most ML models are programmed on relatively small datasets by companies. An important step forward could be taken if data sharing would take place between all institutions.

Important aspects to discuss are the legal and ethical concerns. According to Naik et al., four major issues must be considered in this context: (1) informed consent to use the data, (2) safety and transparency, (3) algorithmic fairness and biases, and (4) data privacy. Transparency of the algorithm can be difficult to achieve because modern computing approaches can deliberately hide the thinking behind the output. To confront that problem, diverse and inclusive programming groups as well as frequent audits of the algorithm should be implemented. The data on which ML models are trained can be subject to underlying bias leading to biased ML models. For that, it is important to acquire good-quality unbiased data. Since technologists are not obligated by law to be held accountable for their actions to the same degree as doctors, the question of whether, and if yes how much, they should take responsibility for the AI system they created, remains. An effort on clear guidelines for the legal and ethical issues concerning AI should be made [85].

## 4. Conclusions

Although AI in medicine is a rapidly growing field that shows promising results, clinical implementation is still lacking. Different studies develop and test their own ML algorithms without open-source access, leading to studies not being reproducible. Finding a clear structure for study design and the implementation of data sharing, as well as the regular use of external validation should be aimed for, in order to drive progress forward.

## Figures and Tables

**Figure 1 diagnostics-12-02235-f001:**
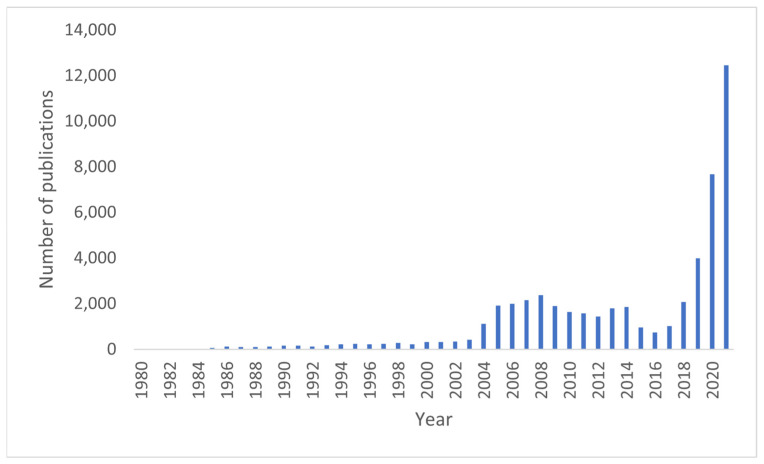
Number of PubMed search results for “artificial intelligence” with an exponential growth in numbers of published articles after 2016.

**Figure 2 diagnostics-12-02235-f002:**
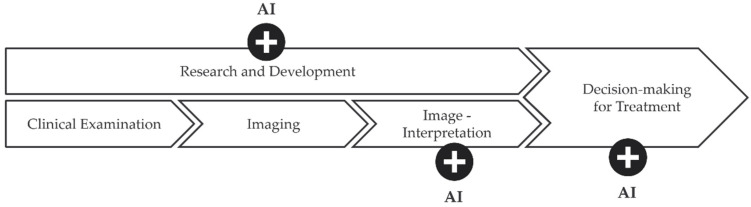
A schematic illustration of the role of artificial intelligence (AI) in orthopedic radiography analysis. AI can play a role in supporting and enhancing image interpretation, research and development, as well as decision-making for treatment plans.

**Figure 3 diagnostics-12-02235-f003:**
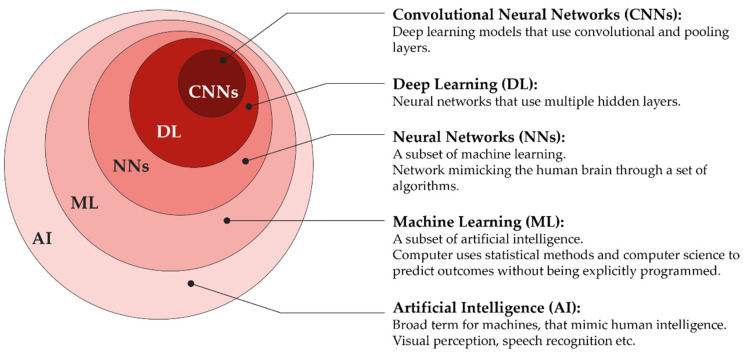
Set diagram describing the relationship of Artificial intelligence (AI) being the umbrella term for everything inside the outer circle, Machine learning (ML) for everything inside the second-outer circle, etc. Neural networks (NN), deep learning (DL), and convolutional neural networks (CNNs).

**Figure 4 diagnostics-12-02235-f004:**
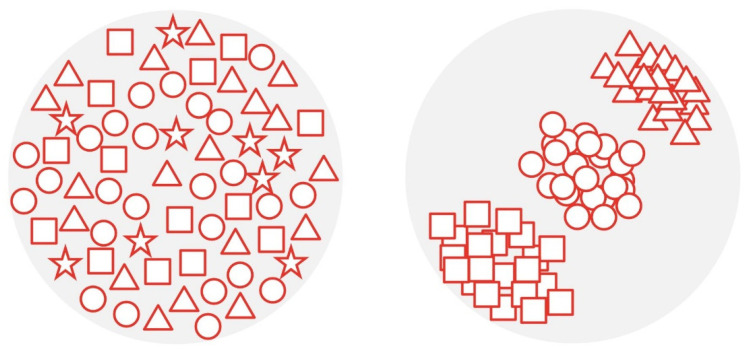
Illustration describing the mechanism of clusterization: In the left image, no specific order can be seen. For clusterization, different shapes are recognized and organized in a group (cluster) as seen in the right image. Shapes that do not belong in any group (e.g., stars) are sorted out. This can help ML models to recognize patterns that would otherwise be difficult to recognize.

**Figure 5 diagnostics-12-02235-f005:**
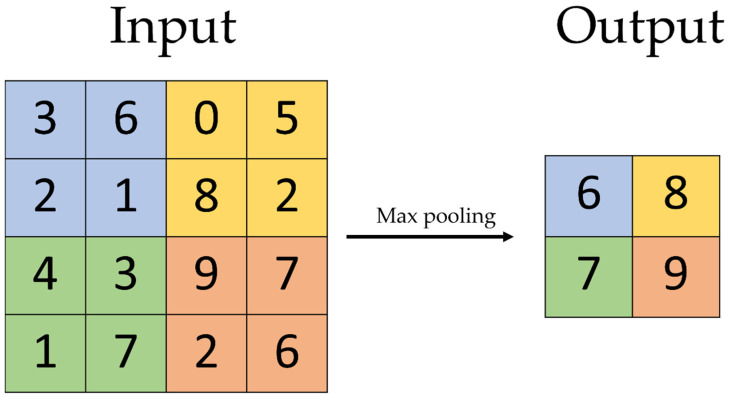
Max pooling layer: the input layer (left) is filtered by a 2 × 2 max pooling layer computing the output with the highest activation (number) of the filtered area in the preceding layer into a smaller output layer. One can imagine a 2 × 2 filter starting on the top left: The first filtered area will be the blue area. The number six, being the highest, will be computed as 1 × 1 in the output, representing the 2 × 2 input. After that, the filter will go to the next area that has not been filtered, e.g., two cells to the right, and repeat the process. With filtering methods like this, the required computational power can be reduced.

**Figure 6 diagnostics-12-02235-f006:**
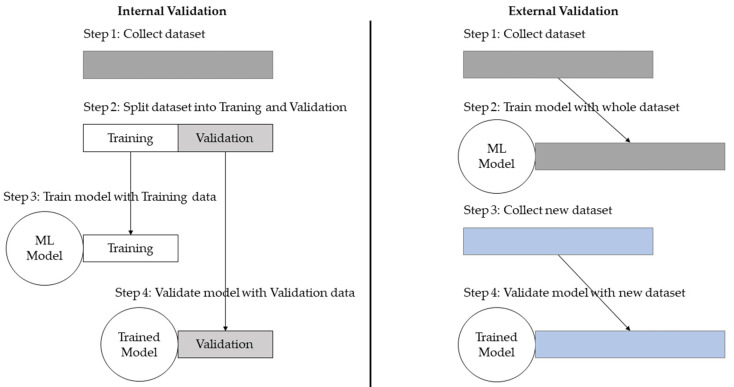
Internal vs. external validation. Internal validation uses one dataset and splits it into training and validation, the ML model is then trained and validated on the same dataset resulting in higher risk of overfitting. External validation uses one dataset for training and a meaningfully different dataset for validation. When (geographically) different datasets are used, the risk of overfitting can be significantly reduced.

## Data Availability

Not applicable.

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
