# Peer review of "Artificial Intelligence in Orthopedic Radiography Analysis: A Narrative Review"

_diagnostics, 2022, doi:10.3390/diagnostics12092235_

Round 1

Reviewer 1 Report

The review presented by the authors is well organized and of great interest. However, in my opinion, it lacks an important aspect:

Artificial intelligence (AI) models and procedures are potentially able to discover hidden, non-obvious clinical patterns in data. However, in my opinion, should be emphasized in the manuscript that to obtain truly useful results in clinical radiological diagnostics, the input data of the ML and AI algorithms must be of good quality.

For example, unfortunately, some ML papers have been published in the literature over the years that had poor-quality MRI images as input data. This implies that the ML algorithm arrives at conclusions that do not adhere to the radiological clinical problem but, for example, reaches conclusions strongly dependent on the poor signal-to-noise ratio of the MRI images. This should be indicated by the authors as a disadvantage of the ML and AI methods, alongside the advantages already listed in the review.

This problem is particularly felt when using MRI for diagnostics of the musculoskeletal system from which to extract input data for algorithms. In fact, unlike soft tissues (for example white and gray matter in the brain), MRI of musculoskeletal tissue is sensitive to artifacts due to the magnetic susceptibility difference between bone and soft tissues, to movement artifacts, and to poor signal-to-noise ratio in poorly free water sites (e.g. nerves, tendons, bone…). 

In this regard, it should also be considered that thanks to the technological evolution of MRI this technology is also entering as a radiological technique in the diagnosis of dental, oral, endodontic, and orthodontic problems. However, due to the problems mentioned above, MRI images although potentially very informative are often not of good quality (see refs. 1- S. Capuani, et al. Nuclear magnetic resonance microimaging for the qualitative assessment of root canal treatment: An ex vivo preliminary study, Diagnostics, 2021;11(6):1012. 2- Niraj, L.K. et al. MRI in Dentistry- A Future Towards Radiation Free Imaging—Systematic Review. J. Clin. Diagn Res. 201610, ZE14–ZE19. 3- Gaudino, C. et al. MR-Imaging of teeth and periodontal apparatus: An experimental study comparing high-resolution MRI with MDCT and CBCT. Eur. Radiol. 201121, 2575–2583.)

Given the close correlation that sometimes exists between orthopedic orthodontics and dental problems, it is also necessary to consider in the review the multiple attempts that have been made to try to solve orthodontic problems with ML and AI (see refs. 4- T. Gili et al. Complexity and data mining in dental research: A network medicine perspective on interceptive orthodontics Orthodontics & Craniofacial Research. 2021; 24, 16-25. 5- Li, P.; et al. Orthodontic Treatment Planning based on Artificial Neural Networks. Sci. Rep. 20199, 2037.)

Author Response

Dear reviewer,

Thank you very much for the insightful comments on our manuscript.

The importance of good quality data has been emphasized again: As mentioned before, overfitting is a challenge for ML algorithms. The size and quality of data, especially with MRI images where the signal-noise-ratio can be worse, determine and influence the performance quality. Although MRI images show a lot of information, they also have more noise (Lines 282-286). We also would like to mention, that importance of quality of data has been discussed when talking about overfitting (Lines 80-86) and internal and external validation (Lines 304-311).

The relationship between orthopedics and orthodontics has been added and the potential of MRI in orthodontics is discussed: Research has shown that orthopedic problems can also be related to orthodontics. Neural networks have produced good performance in treatment planning and can offer guidance for less experienced orthodontists. For example, malocclusion has been shown to be related to scoliosis and weak body posture. The implementation of MRI in orthodontics has shown great potential for radiation free imaging. Compared to the currently used cone beam CT, MRI has shown better results in detection of periodontal structures, soft tissue debris and unfilled spaces in radicular canals, which is necessary information for endodontic treatment. Additionally, orthodontic treatment is mostly received during childhood, which makes MRI even more desirable (Lines 274-282).

We hope that the added revisions correspond to your expectations.

Best regards,

Kenneth Chen on behalf of the co-authors

Reviewer 2 Report

  • Please find it in the attachment.

Author Response

Dear reviewer,

Thank you very much for the insightful comments on our manuscript.

We added sections that call for caution when using AI in bone age assessment as well as leg alignment measurements:

It is important, however, that caution is taken when assessing bone age in patients with skeletal disorders like Hutchinson-Gilford progeria syndrome, where patients show a more advanced rate of skeletal maturation. Large datasets, which are needed to train ML models, are much more difficult to acquire for rare diseases, which makes experience and supervision by humans a necessity. (Lines 219-223)

An important note to mention is that common exclusion criteria for studies mentioned above are severe deformities of the lower limb and bad image quality, emphasizing again that experienced clinicians are indispensable as supervisors. This becomes even more important in undiagnosed bony abnormalities. (Lines 254-257)

The word “detection” has been replaced by “confirmation”. (Line 265)

The possibility of an adverse effect by AI has been added to the discussion:

On the other hand, we cannot rule out the possibility, that medical students and junior doctors could rely too much on AI, which can become a hindrance in developing analytical and observational skills as well as to learning the process of decision making. (Lines 358-360)

We hope that the added revisions correspond to your expectations.

Best regards,

Kenneth Chen on behalf of the co-authors

Reviewer 3 Report

Following are the suggestions to authors to improvise the manuscript

1. The abstract should clearly reflect the objective of the review.

2. Authors should follow the submission guidelines of the journal and accordingly revise the citation style and referencing style.

3. The authors need to in detail explain the plots/graphs highlighting the critical observations obtained and what is the outcome.

4. Figures/ graphs should be replotted with the relevant tool and provide high-resolution images

5. Kindly check if all the references are cited in the manuscript and vice versa

6. Avoid using citation to the subheading or to the caption of the figures and tables instead cite the reference in the text of the paragraph

7. All the abbreviations used in the manuscript and the figures, in full form should be present at least once in the manuscript text.

8. Ther review should include at least five or more latest and highly cited articles.

9. The Conclusion section is missing in the manuscript. Kinly provide the overall concluding remarks for the review

10. Kindly check if the figures have copyrights and Fig 1 can be reported with plain white background

11. Kindly include a schematic diagram in the introduction section to provide an overview of intervention of Artificial Intelligence in Orthopedic Radiography Analysis. This will help the readers to understand the role and importance of AI in the specific domain.

12. The introduction section is weak. Kindly refer to the articles to improvise the section. 

https://doi.org/10.3390/jcm10091864

13. Authors can discuss legal and ethical concerns of adoption of AI in the clinical practice

https://doi.org/10.3389/fsurg.2022.862322

Author Response

Dear reviewer,

Thank you very much for the insightful comments on our manuscript.

  1. The abstract has been modified, so that the objective becomes clearer.
  2. We have adjusted the citation and reference style according to the instructions for authors of MDPI.
  3. Descriptions of plots and diagrams have been modified for more detailed explanation
  4. Images have been replotted and fulfill the required pixel size according to instructions for authors of MDPI. PNG files can be found in the attachments.
  5. References have been rechecked for citation in the manuscript and vice versa, missing references have been added.
  6. Tables and figures have been rechecked, no citations were found in subheadings or descriptions.
  7. We made sure to include all abbreviations in full form at least once in the manuscript.
  8. Highly cited articles:
  9. Hamet, P.; Tremblay, J. Artificial intelligence in medicine. Metabolism 2017, 69S, S36-S40, doi:10.1016/j.metabol.2017.01.011
  10. Gillies, R.J.; Kinahan, P.E.; Hricak, H. Radiomics: Images Are More than Pictures, They Are Data. Radiology 2016, 278, 563–577, doi:10.1148/radiol.2015151169.
  11. Murdoch, T.B.; Detsky, A.S. The inevitable application of big data to health care. JAMA 2013, 309, 1351–1352, doi:10.1001/jama.2013.393.
  12. Litjens, G.; Kooi, T.; Bejnordi, B.E.; Setio, A.A.A.; Ciompi, F.; Ghafoorian, M.; van der Laak, J.A.W.M.; van Ginneken, B.; Sánchez, C.I. A survey on deep learning in medical image analysis. Image Anal. 2017, 42, 60–88, doi:10.1016/j.media.2017.07.005.
  13. Shen, D.; Wu, G.; Suk, H.-I. Deep Learning in Medical Image Analysis. Rev. Biomed. Eng. 2017, 19, 221–248, doi:10.1146/annurev-bioeng-071516-044442.

  1. Conclusion section has been added to the manuscript
  2. All tables and figures have been drawn by Kenneth Chen, no copyright provisions were violated, and Figure 1 has been replotted with plain white background
  3. A schematic illustration of the role of AI in orthopedic radiography analysis has been added.
  4. The introduction section has been modified.
  5. A section discussing legal and ethical concerns has been added to the discussion section

We hope that the added revisions correspond to your expectations.

Best regards,

Kenneth Chen on behalf of the co-authors

Round 2

Reviewer 2 Report

The additions are persuasive 

Author Response

Dear reviewer, 

Thank you very much for your response.

Best regards,

Kenneth Chen on behalf of the co-authors

Reviewer 3 Report

The authors have addressed the suggestions and have improvised the manuscript to best possible extent. However, I suggest one additional change to be made by authors is with respect to Fig. 3. I would suggest authors to revise the figure and present a better image which would clearly classify the levels of AI.

In addition authors can consider to add an Schematic or Graphical abstract in introduction section with reference to AI in Orthopedic Radiography that would provide readers with a overview of the Role of AI.

Author Response

Dear reviewer,

Thank you very much for your response.

Figure 3 has been redrawn and added to the manuscript. We believe that the levels of AI are now clearer. 

The PNG-file will also be added in the resubmission.

A graphical abstract has been drawn and will be submitted under graphical abstract.

We hope that the added revisions correspond to your expectations.

Best regards,

Kenneth Chen on behalf of the co-authors